# Ultra-Processed Food Consumption and Its Association with Risk of Obesity, Sedentary Behaviors, and Well-Being in Adolescents

**DOI:** 10.3390/nu16223827

**Published:** 2024-11-08

**Authors:** Aristides M. Machado-Rodrigues, Cristina Padez, Daniela Rodrigues, Elizabete A. Dos Santos, Liliana C. Baptista, Margarida Liz Martins, Helder Miguel Fernandes

**Affiliations:** 1University of Coimbra, Faculty of Sport Sciences and Physical Education, 3040-248 Coimbra, Portugal; lbaptista@fcdef.uc.pt; 2Research Centre for Anthropology and Health, University of Coimbra, 3001-401 Coimbra, Portugal; cpadez@antrop.uc.pt (C.P.); rodrigues1323@gmail.com (D.R.); 3Department of Life Sciences, University of Portugal, 3000-456 Coimbra, Portugal; 4Nutrition Department, São Paulo State University, São Paulo 01246-904, Brazil; elizabete.nutri21@gmail.com; 5Coimbra Health School, Instituto Politécnico de Coimbra, 3045-043 Coimbra, Portugal; margarida.liz@estesc.ipc.pt; 6Polytechnic Institute of Guarda, 6300-559 Guarda, Portugal; hmfernandes@ipg.pt; 7Sport Physical Activity and Health Research & INnovationCenTer (SPRINT), 6300-559 Guarda, Portugal

**Keywords:** ultra-processed food, obesity, well-being, sedentary behaviors, adolescence

## Abstract

Background/Objectives: The literature on consumption of ultra-processed food (UPF) using the NOVA classification is still limited. Therefore, the aim of the present study was twofold: (i) to compare the UPF consumption, sedentary behaviors, and well-being perception between boys and girls; and (ii) to investigate the association between the UPF consumption and risk of overweight, sedentary behaviors, and well-being in adolescents. Methods: The present cross-sectional study comprised a sample of 245 adolescents (131 boys) aged 12–17 years-old (*M* = 14.20; *SD* = 1.09). Height and weight were assessed, and subsequently, the BMI was computed; furthermore, total body fat percentage was measured with bioelectrical impedance. Daily consumption of UPF was assessed by the NOVA screener and time spent sedentary was assessed by the Midlands Behavior Health 2024 questionnaire. The Mental Health Continuum-Short Form (MHC-SF) was used to measure adolescents’ psychosocial well-being. Pearson’s correlations and logistic regression analysis were used, controlling for biological, behavioral, and socio-economic confounders. Results: No sex differences were observed for the different UPF NOVA subscales. Boys reported higher computer use levels than girls on the weekend (*p* = 0.025), and they spent more time playing electronic games during the week (*p* = 0.005) and on the weekend than their female counterparts (*p* < 0.001). Moreover, boys reported higher scores in all well-being dimensions (*p* < 0.001) than girls. Conclusions: The findings revealed, after controlling for sex, sedentary time, and active behaviors, adolescents who consumed UPF on the previous day tended to be associated with a higher risk of being overweight, but also marginally without statistical significance (OR = 0.91, 95% CI: 0.83–1.01, *p* = 0.06). Of relevance, the present study revealed that both boys and girls of mothers with high educational levels were less likely to be classified as overweight or obese youth.

## 1. Introduction

Pediatric obesity is an increasingly concerning public health issue requiring a comprehensive examination of dietary behaviors and sedentary practices associated with this condition, particularly examining specific geographic settings. The consumption of ultra-processed foods is rising among adolescents [1,2] as result of their widespread availability and high tastiness and sweetness, and this, combined with a sedentary lifestyle, substantially contributes to the pediatric obesity epidemic. These behaviors have been particularly prevalent in Western countries [3] and Southern Europe [4], where the prevalence of adolescents at risk of obesity is among the highest worldwide.

Changes in the dietary patterns of populations in recent decades have determined an increase in consumption of “convenience” foods and beverages, ready-to-eat products, and packaged food, which have replaced traditional dietary patterns based on in natura, minimally processed foods and freshly prepared meals. On the other hand, the dietary assessment paradigm has also changed over the last decades; it traditionally relied on 24 h dietary recalls of healthy nutritional items (i.e., fruit, soup, etc.) and, subsequently, on questionnaires focused on sugar-sweetened beverage consumption, one of the most detrimental nutritional habits among adolescents [5,6]. It is now evident that a more detailed exploration of the consumption of ultra-processed foods—encompassing both solid products, such as sugary and salty snacks, as well as sugary and soft drinks—is essential. Therefore, it is crucial to develop scientifically valid and reliable instruments that also consider geographical contextualization, enabling an adaptation to the dietary patterns of different populations. This enhanced approach can yield more accurate and relevant data for epidemiological studies, eventually leading to the development of more effective interventions aimed at addressing the problem of obesity.

Recent studies have increasingly highlighted the association between the consumption of ultra-processed foods (UPF) and the rising rates of obesity among adolescents [7]. Ultra-processed foods are typically high in added sugars, unhealthy fats, sodium, and chemical additives such as dyes, preservatives, flavorings, flavor enhancers, while being low in essential nutrients and fiber. This dietary pattern is linked not only to excessive caloric intake, but also to unfavorable metabolic outcomes. Research shows that adolescents consuming a higher proportion of UPFs tend to exhibit increased body mass index (BMI), along with elevated levels of inflammatory markers and metabolic syndrome components [8,9]. Furthermore, UPFs often contribute to poor dietary habits, reinforcing a cycle of unhealthy eating that increases the risk of obesity and related comorbidities. The effect of ultra-processed foods on adolescent obesity can also be seen in the context of behavioral and psychological factors. The convenience and sweetness of UPF are significant contributors to their popularity among young individuals, often leading to overconsumption and displacement of healthier food choices. Studies suggest that reliance on UPFs can influence eating behaviors, facilitating habits such as mindless snacking and emotional eating [10,11]; among southern European adolescents, UPF was also associated with several psychosocial factors and eating behaviors, suggesting that increased adherence to the Mediterranean diet would possibly result in lower UPF consumption [12]. As a result, health professionals and policymakers are increasingly advocating for interventions aimed at reducing UPF consumption among adolescents, promoting the consumption of whole, in natura, minimally processed foods to support healthier dietary patterns and mitigate obesity risk.

In countries with high rates of pediatric obesity, there is an emerging need for a more precise assessment that can clearly elucidate the true impact of dietary habits on youth obesity. In Portugal, the lack of studies in this area has prompted the development of instruments to evaluate the consumption habits of ultra-processed foods, which is currently in the final stages of validation, following a similar extensive process of transcultural contextualization [13]. Another source of variation which has received less attention is the association between ultra-processed food and well-being among adolescents. Recent scientific studies and systematic reviews within the last five years have shown bidirectional associations between ultra-processed food consumption and adverse mental health outcomes, particularly symptoms of common mental disorders like depression and anxiety [10,14]. These studies suggest that greater exposure to ultra-processed foods is linked to a higher risk of adverse health outcomes, including mental health issues. Further research is needed to confirm these findings and to explore the effects of ultra-processed food consumption on mental well-being in adolescents.

Given the limited available information about the relationship between these constructs in Southern European communities, the present study achieves greater insight, since youth from those countries have the highest prevalence of overweight/obesity. In addition, there is an age-related decline in active behaviors throughout the life span, and the transition from childhood to adolescence seems to be a critical period to be studied. To the best of our knowledge, there are no studies that have analyzed the afore-mentioned constructs together and/or combined (i.e., UPF and risk of obesity, sedentary time in its different forms, and even the different dimensions of well-being) among Portuguese adolescents. Therefore, the aim of the present study was twofold: (i) to compare the consumption of ultra-processed foods, sedentary behaviors, and well-being perception between boys and girls; and (ii) to investigate the association between the consumption of ultra-processed foods, body fat, sedentary behaviors, and well-being in adolescents. It was hypothesized that females would spend more time engaged in sedentary activities than boys, and that there would not be differences in the consumption of ultra-processed foods by sex; furthermore, it was hypothesized that the consumption of ultra-processed foods would be positively related to the risk of being overweight, as well as associated with less perception of well-being among adolescents, particularly from parents/families with lower educational levels.

## 2. Materials and Methods

### 2.1. Participants and Study Design

Participants were recruited under the scope of the project PMBH-2024 “Portuguese Midlands Behavioral Health”, aiming to assess the behavioral lifestyle and health markers of Portuguese youth and its association with pediatric obesity. The sampling design for the present cross-sectional study conducted in 2024 was the same, as it is an extension of the previous projects carried out by our scientific team in 2016–2017 (Portuguese Prevalence Study of Obesity in Childhood (PPSOC)) to assess childhood obesity prevalence and the obesogenic environment [15]. Thus, using similar methodological/instrumental procedures, a total of 245 adolescents (114 girls and 131 boys) aged 12–17 years old (*M* = 14.20; *SD* = 1.09) from two random selected public high schools of the Coimbra (*n* = 101) and Viseu (*n* = 144) districts were recruited to participate in the present study. The sample age range is coincident with the scholarly level or academic years studied at the same degree; since students in elementary school were studied (Portuguese 3rd cycle of the elementary studies), participants were aged from 12 to 17 years of age. According to the literature [16], once gender was commonly indicated as a factor affecting physical activity, well-being, and weight status across adolescence, the analyses were performed separately for male and female adolescents.

This study was conducted according to the Declaration of Helsinki and approved by the ethics committee of the Portuguese Directorate-General for Education (Study Registration Nº 1472300001/MIME). In addition, the head of each aforementioned school provided ethical approval, which also required informed assent from each participant and their parents/guardians. Prior to data collection, informed written assent was obtained from parents or guardians.

### 2.2. Anthropometry

Height and weight were assessed for each participant according to standardized procedures. Weight was measured to the nearest 0.1 kg using a calibrated scale (Tanita UM076, Tokyo, Japan), and height was measured to the nearest cm using a wall-mounted rule. The body mass index was computed as weight (in kilograms) divided by height (in meters) squared. The BMI was calculated, and youth were classified as normal weight, overweight, or obese using age-and sex-specific BMI cut-offs recommended by the International Obesity Task Force [17]. The total body fat percentage was measured with bioelectrical impedance analysis using the Tanita UM076. After entering participants’ characteristics (age, sex, and height), participants were instructed to stand barefoot on the platform footplates, according to the manufacturer’s guidelines.

### 2.3. Ultra-Processed Food Consumption

The NOVA-UPF screener was used for UPF consumption assessment in adolescents. The Nova-UPF screener was developed in Brazil [18], and it is a short food-based questionnaire that specifically addresses the consumption of UPFs on the previous day through “yes” or “no” questions. The Nova-UPF screener was adapted for the Portuguese adolescent population (data not yet published), assigning a score ranging from 0 to 28 according to a list of 28 predefined UPF subcategories within 3 major groups (beverages, products that replace or accompany meals, and products often consumed as snacks). The score obtained with this tool accurately reflects the level of energy intake from UPFs and identifies which are the main dietary sources of UPFs for children and adolescents. This tool is simple and quick to apply and requires a low workload, making it feasible to quickly quantify the consumption of UPFs and the relation to overweight and obesity, as well as other health outcomes.

### 2.4. Sedentary Behaviors

Leisure-time sedentary behaviors were self-reported. First, participants were asked about the total sedentary time during a typical weekday and weekend day, and it was expressed separately by the usual number of minutes (on average) spent sitting (reading, writing, studying, talking, watching TV, or using a PC or smartphone, separately). Next, participants were asked to indicate the sedentary time (in minutes) during a typical weekday and weekend day for the following specific behaviors: watching TV, using a PC, playing electronic games, using a smartphone, using tablets, studying at home, other activities (such as board games or reading), studying at school during recess, or using electronic devices (smartphones, tablets, etc.) during recess.

### 2.5. Well-Being

A validated Portuguese version of the Mental Health Continuum—Short Form (MHC-SF) was used to measure adolescents’ psychosocial well-being [19]. The MHC-SF includes 14 items grouped into three dimensions: (i) emotional well-being (3 items), (ii) social well-being (5 items), and (iii) psychological well-being (six items). These items were rated on a six-point Likert scale ranging from 0 (never) to 5 (every day), depending on the frequency of well-being symptoms in the last month. The total scores of the three dimensions were computed as the sum of the scores of the items included. Omega reliability values showed adequate internal consistency for all dimensions, namely, emotional well-being (ω = 0.83), social well-being (ω = 0.84), and psychological well-being (ω = 0.81).

The physical well-being was assessed using the respective subscale of the Portuguese version of the KIDSCREEN-27 questionnaire. This dimension includes five items rated on a five-point Likert scale, which are summed to provide a total score, with higher scores indicating better physical well-being. Internal consistency analysis revealed high omega reliability (ω = 0.88).

### 2.6. Parental Education

The educational backgrounds of the fathers and mothers were used as a proxy for socio-economic status. This was based on the Portuguese Educational System ((1) 9 years or less—sub-secondary; (2) 10–12 years—secondary, and (3) higher education)). The three educational levels were defined, respectively, as: 1 = low education; 2 = middle education; and 3 = high education. Similar procedures have been used in the Portuguese context [20].

### 2.7. Statistical Analyses

Preliminary inspection of the data was conducted for accuracy and missing values. The missing value analysis indicated 0.78% of missing data that appeared to be missing completely at random, as suggested by Little’s test: χ^2^_(627)_ = 670.08, *p* = 0.113. Therefore, missing data were imputed using the expectation-maximization algorithm based on all available data, since it has previously been demonstrated to be an effective technique for small to moderate sample sizes [21]. Descriptive statistics such as the means (*M*), standard deviations (*SD*), and frequency and percentage (%) were calculated, as well as independent t-tests and Pearson’s or point-serial bivariate correlations between variables under analysis. The subscale’s internal consistency (reliability) was evaluated using McDonald’s omega coefficients.

Furthermore, associations between UPF consumption (i.e., the UPF score was analyzed as a continuous variable following the original procedure) and the overweight/obesity risk, controlling for potentially confounding effects of sex, chronological age, SB (i.e., time spent using different screen media devices (minutes/day), such as TV viewing, PC, electronic games, iPads/Tablets, and smartphones, separately on weekdays and on the weekend), parental BMI, and parental education, were estimated using logistic regression analysis. In the minimally adjusted model (Model 1), UPF was the sole predictor of the adolescent’s overweight/obesity risk. Sex and chronological age were subsequently added as potential confounders (Model 2). Furthermore, SB (i.e., TV viewing, PC, electronic games, iPads/Tablets, smartphone, separately) was added as a potential confounding factor (Model 3). In addition, parental BMI was included in the subsequent model as a potential confounding factor (Model 4). Finally, parental education was included in the final model as a potential confounding factor (Model 5).

Due to the moderate sample size (*n* < 250), bootstrap resampling methods [22] were also used to estimate bias-corrected and accelerated (BCa) 95% confidence intervals for the mean differences between sex groups and for the Pearson’s correlation coefficients. Robust estimates were obtained based on 5000 bootstrap samples (random number generator: Mersenne Twister, seed 151024).

All analyses were conducted using IBM SPSS Statistics for Windows, version 27 (IBM Corp., Armonk, NY, USA). The level of statistical significance was set at 5%.

## 3. Results

Descriptive statistics are presented for the total sample and sex differences in Table 1. The mean age of the girls was 14.2 (±1.2) years, and for boys, it was 14.2 (±1.0) years. About 17.5% of girls were categorized as overweight and 7.9% as obese; the corresponding values for boys were 15.3% and 3.1% for overweight and obese males, respectively.

The results for the total sample revealed similar UPF consumption levels in the three food subcategories of the NOVA screener, obtained neither from the 24 h recall nor from outside-the-home consumption. During weekdays, adolescents reported higher use of smartphones, followed by studying and using a PC. The most reported sedentary behaviors at the weekend were using a smartphone or a PC and watching TV.

No sex differences were observed for the different UPF subcategories. As expected, girls showed higher BMI levels (*p* < 0.001) and body fat percentages (*p* < 0.001) than boys. Boys reported higher PC use levels than girls at the weekend (*p* = 0.025), and they spent more time playing electronic games during the week (*p* = 0.005) and on the weekend than their female counterparts (*p* < 0.001). On the other hand, girls reported more time spent studying during the week (*p* = 0.006) and at the weekend (*p* = 0.007), as well as more time spent engaged in other activities at the weekend, such as board games or reading (*p* = 0.026), than their male peers. Moreover, boys reported higher scores in all well-being dimensions (*p* < 0.001) than girls. The analysis of the additional bootstrap estimates showed a similar trend of results (see Appendix A).

Correlation results between the main study variables (Table 2) revealed that the 24 h recall consumption of UPFs, including sugary drinks and yogurts, was positively related to watching TV (weekend), playing electronic games, and using a smartphone (weekdays). Conversely, it was negatively associated with body fat percentage and time spent studying during weekdays and at the weekend. The consumption of UPFs such as sugary drinks and yogurts away from home was positively associated with the total time spent sedentary at the weekend and negatively related to body fat percentage and time spent studying at the weekend. The analysis of the additional bootstrap estimates showed a similar trend of results (see Appendix A).

A similar trend of associations was observed for the 24 h recall consumption of UPFs such as packaged and fast food, except for the additional positive correlations with watching TV on weekdays and using a smartphone at the weekend. The consumption of UPFs such as packaged and fast food away from home was only negatively related to body mass index. The consumption of UPFs such as sweet and salty snacks outside the home was negatively and significantly associated with body fat percentage. Moreover, the reported consumption of UPFs was not significantly related to any of the well-being dimensions for adolescents.

Associations between UPF consumption and overweight/obesity risk, controlling for the aforementioned confounding effects, are presented in Table 3 (UPF—last day) and Table 4 (UPF—away from home). After controlling for all potential confounders, we found that UPFs tend to be associated with the risk of being overweight on the last day (OR = 0.91, 95% CI: 0.82–1.01, *p* = 0.08) and away from home (OR = 0.93, 95% CI: 0.84–1.02, *p* = 0.09); notably, after controlling for sedentary and active behaviors (i.e., Model 3), adolescents who consumed UPF on the previous day tended to be associated with an higher risk of being overweight, but also marginally, without statistical significance (OR = 0.91, 95% CI: 0.83–1.01, *p* = 0.06).

The final regression model revealed that adolescents of mothers with high educational levels were less likely to be classified as overweight or obese youth (OR = 0.83, 95% CI 0.70–0.98, *p* = 0.02); furthermore, those adolescents who spent more time using PCs at the weekend were more likely to be classified as overweight (OR = 0.99, 95% CI 0.98–1.00, *p* = 0.04).

## 4. Discussion

Pediatric obesity remains an important challenge in global health, since the number of youths aged 5 to 19 years with obesity is predicted to rise to 254 million by 2030. Understanding specific nutritional factors that may influence adolescents’ obesity/adiposity should be clearly analyzed, especially in under-studied populations of Southern Europe where the highest rates of obesity are prevalent (i.e., Portuguese adolescents). In addition, there is an age-related increase in sedentary behaviors throughout the life span, and adolescence seems to be a critical period to be studied. The findings of the present study reveal there are no differences between boys and girls in UPF consumption, which is consistent with the Avon Longitudinal Study of Parents and Children (ALSPAC) in the UK [23] and the USA [24], but contrasts with adolescents from Bangladesh [25], Brazil [26], the UK [27], and Spain [10]. On the other hand, the present study reveals, after controlling for sex, sedentary time, and active behaviors, that children who consume UPFs tend to be associated with a higher risk of being overweight, but marginally, without statistical significance (*p* = 0.06), highlighting that the risk of being overweight could be also related to other behavioral factors, as well as environmental and socio-cultural features of the regions where adolescents and their families are living.

To our knowledge, this is one of the first studies that has investigated the combined association between UPF consumption, several behavioral and emotional factors, and the risk of overweight among Portuguese adolescents. The main findings of the present study are in line with Brazilian studies of adolescents [28] in which no associations were observed between the consumption of UPFs and anthropometric indicators (i.e., BMI, waist circumference, and waist-to-hip ratio). However, contrasting results were observed in a recently published systematic review [29] emphasizing a positive association of UPF consumption with childhood overweight/obesity. Despite these controversial results in the literature, the present study also points out an interesting relationship observed between UPF consumption and increased sedentary time, a well-established risk factor for obesity, highlighting the importance of the presented innovative data and findings and confirming the complex etiology of pediatric obesity.

Excessive screen time is commonly related to other concerning and significant lifestyle changes, such as dietary habits [30,31]. The present study revealed that UPF consumption was positively related to watching TV (weekends), playing electronic games, and using smartphones on weekdays. In addition, correlation analyses showed that consumption of ultra-processed foods and beverages was associated with increased time spent engaged in sedentary activities, such as watching TV and using smartphones, and reduced dedication to studies, especially on weekends. In a nationally representative study of US youth, excessive TV viewing (i.e., ≥5 h/day) was associated with daily intake of sugar-sweetened beverages and obesity in both boys and girls [32]. Furthermore, although the present study did not reveal associations between UPF consumption and PC use, adolescents who spent more time using a PC on the weekend were more likely to be classified as overweight, which needs to be clearly considered.

Another source of variation in sedentary time is the sex difference effect identified in the literature [30,31]. The findings of the present study reveal that boys reported higher screen time (PC and electronic games) both at the weekend and during the week. In contrast, girls spent more time studying during the week and on the weekend, and they also reported more time devoted to other activities such as board games or reading. Similar results were observed among Chinese adolescents, revealing that males were 4.55 times more likely to engage in excessive screen time than females [30]. Furthermore, Canadian boys spent more time playing video/PC games than female adolescents who spent more time leisure reading [31], which corroborates the results found in the present study and may explain why girls spend less time on screens, as they are dedicating themselves to other activities such as studying.

Of interest, the present study demonstrated that smartphone use was the most reported sedentary behavior during the week and on weekends. The more adolescents use smartphones, the more exposed they are to social media [33]. In fact, a recent systematic review [34] revealed that adolescents are more likely to recall unhealthy foods with celebrity influence recurring in advertisements. Indeed, if “unhealthy” food and beverages companies use social media to target their product, marketing specifically to adolescent audiences, they are not contributing to healthy lifestyles among young communities. Law No. 30/2019 on restrictions of food marketing through the analysis of food advertising aimed at youth was recently created in Portugal. However, a recent study showed that Portuguese adolescents are still potentially exposed to many advertisements for unhealthy foods (i.e., chocolate and bakery products (42.0%), soft drinks (26.7%), and yogurt (16.0%)) on television despite regulatory and marketing restriction policies [35]. Indeed, UPF consumption and sedentary behaviors are closely linked among adolescents, leading to a range of adverse health outcomes. These behaviors, as well as the low dietary diversity and reduced opportunities for daily physical activity and social interaction, reinforce each other, creating a difficult cycle to break out of without targeted interventions.

Previous studies have shown that socioeconomic level and parental education influence food consumption and the prevalence of obesity in children and adolescents [15,36,37]. In the present study, adolescents of mothers with high educational levels were less likely to be classified as overweight or obese, corroborating results from Italian adolescents that revealed that a higher educational level was also associated with reduced energy from UPFs in the diet [12]. Findings concerning Belgian, Hungarian, Greek, and Spanish adolescents from the HELENA study revealed that a low socioeconomic status (maternal education and occupation) was significantly associated with consuming energy-dense beverages (soft drinks, beer, or fruit juice) while watching TV, but only in girls. The authors pointed out that adolescents with less favorable socioeconomic circumstances run the risk of having unhealthy behaviors which favor the onset of obesity [38]. It is assumed that mothers with higher levels of education have better access to quality food and adequate nutritional information and guidance, which may result in more effective care for their children’s health.

The transition from childhood to adolescence seems to be a critical period to be studied, particularly when evaluating sex-specific issues related to well-being. The present study revealed that boys presented higher scores in all well-being dimensions. Similar results have been found in other studies, where girls reported lower well-being and worse mental health [16,39]. Differences between the instruments used to assess well-being should be considered. However, factors such as earlier puberty, lower life satisfaction, self-confidence, dissatisfaction with body image, and having to deal with gender-related norms may justify these differences [16,40]. On the other hand, UPF consumption was not associated with psychological or physical well-being in the present study. Recent scientific studies and systematic reviews within the last five years have shown bidirectional associations between UPF consumption and adverse mental health outcomes, particularly symptoms of common mental disorders like depression and anxiety [11,14]. These studies suggest that greater exposure to ultra-processed foods is linked to a higher risk of adverse health outcomes, including mental health issues.

Some potential mechanisms which are linked to several adverse effects on health and behaviors might explain this relationship. The literature commonly states these foods can impact the brain’s reward system, leading to addictive-like behaviors characterized by intense cravings, loss of control, and continued consumption despite adverse outcomes [41,42]. UPFs may also disrupt normal appetite regulation, resulting in overeating and unhealthy eating patterns, and can cause nutritional deficiencies detrimental to brain development and function [40,43]. Moreover, UPF consumption can alter the gut–brain axis by affecting the gut microbiome, which interacts with the brain through neural, inflammatory, and hormonal pathways [40]. The ingredients of UPFs may promote inflammation, which is usually associated with mental health issues like depression and anxiety [43,44]. It should also be noted that UPF consumption may disrupt circadian rhythms due to irregular eating patterns, and socially, it is often linked to reduced family meals, decreased intake of whole foods, and fewer opportunities to develop healthy eating habits and cooking skills [40]. Despite the fact that these mechanisms provide plausible explanations for the impact of ultra-processed foods on adolescent well-being, the present study of Portuguese adolescents with high rates of overweight and excessive screen time did not reveal significant associations between UPF consumption and any well-being dimensions; therefore, there is a crucial need for further studies to fully understand the complex interactions between diet, neurobiology, and mental health in this age group. The long-term effects and the potential for reversibility of these impacts also require urgent research.

Combined analysis of several factors and different dimensions associated with pediatric overweight in the present study emphasized the need to avoid high levels of sedentary time and unhealthy food consumption, pointing out the educational role of the familial setting to potentiate strategies promoting active and healthier lifestyles among adolescents. Furthermore, the results of the current study also highlight the need to encourage the improvement of healthy nutritional habits in adolescents in the school setting. For example, among the potential interventions to educate about and promote healthy eating habits are the construction and cultivation of school gardens, the development of recipes with students to use fresh produce from the garden while excluding ultra-processed ingredients, and the organization of healthy recipe competitions. Additionally, it is essential to implement actions and playful activities that encourage the availability of nutritious foods and promote healthy choices in school cafeterias, using tools such as the Food Wheel, the Mediterranean Diet Pyramid, and the NOVA classification as references.

The major strengths of this study are the use of an innovative method to assess ultra-processed food in Portuguese adolescents, and particularly the demographic features of those Southern European participants with high rates of overweight. Furthermore, the combined analysis of potential factors from different aspects (i.e., biological, behavioral, and emotional) and the diverse etiology associated with UPFs should be clearly highlighted. The limitations of the present study should also be recognized; firstly, this study has a cross-sectional design and, therefore, it is not possible to infer causal relationships. Furthermore, the sample of the present study is not representative of the Portuguese adolescent population, which could be an additional limitation; in fact, even after the additional robust estimates of 5000 bootstrap samples findings showed a similar trend of results (see Appendix A), other specific socio-cultural features of the communities where data were collected (i.e., from non-urban setting) might have had an impact on the magnitude of association between the studied variables. On the other hand, food habits (i.e., UPFs), sedentary behavior, and active behaviors were assessed by self-report instruments, which might be inaccurate, as they are reliant on accurate recall. Finally, the information on how much ultra-processed foods contributed in terms of total daily energy to the adolescents’ diets was not assessed; although it was not one of the main purposes of the present study, it might contribute to obtaining a better picture of the impact of UPF consumption on adolescents’ lifestyles, which should be considered in future studies. Therefore, further research is needed, not only using objective and more reliable instruments to assess UPFs, physical activity, and sedentary behaviors, but also further studies from different geographic settings, which might include larger and representative samples from different socio-economic backgrounds to confirm or refute the findings of the present study.

## 5. Conclusions

The findings of the present study revealed, after controlling for sex, sedentary time, and active behaviors, that adolescents who consumed UPFs tend to be associated with a higher risk of being overweight, but marginally, without statistical significance (*p* = 0.06). Of relevance, the present study revealed that both boys and girls of mothers with high educational levels were less likely to be classified as overweight or obese youth. Specific strategies are needed for avoiding unhealthy lifestyles that last into adulthood and might increase the growing prevalence of chronic non-communicable diseases in our populations.

## Figures and Tables

**Table 1 nutrients-16-03827-t001:** Descriptive statistics for the total sample and sex differences for the main variables.

Variables	Total Sample (*n* = 245) *M* ± *SD*	Boys (*n* = 131) *M* ± *SD*	Girls (*n* = 114) *M* ± *SD*	*p*
NOVA—UPF 24 h recall				
Drinks and yogurts	2.31 ± 1.74	2.47 ± 1.80	2.13 ± 1.65	0.125
Packaged and fast food	2.69 ± 2.39	2.84 ± 2.52	2.51 ± 2.22	0.280
Sweet and salty snacks	2.36 ± 2.06	2.30 ± 2.09	2.43 ± 2.04	0.608
Total score	7.36 ± 4.86	7.61 ± 4.95	7.07 ± 4.75	0.389
NOVA—UPF away from home				
Drinks and yogurts	1.37 ± 1.75	1.54 ± 1.75	1.18 ± 1.73	0.107
Packaged and fast food	1.55 ± 2.63	1.74 ± 2.87	1.32 ± 2.32	0.214
Sweet and salty snacks	1.40 ± 2.22	1.55 ± 2.48	1.24 ± 1.88	0.273
Total score	4.32 ± 5.96	4.83 ± 6.43	3.74 ± 5.33	0.153
Body mass index (kg/m^2^)	20.79 ± 4.21	19.91 ± 3.75	21.82 ± 4.49	<0.001
Body fat (%)	20.08 ± 8.81	16.22 ± 7.47	24.52 ± 8.14	<0.001
Sedentary time during weekdays				
Total time, min/day	219.55 ± 152.86	209.85 ± 144.98	230.69 ± 161.37	0.288
Watching TV, min/day	60.80 ± 81.55	60.80 ± 90.19	60.79 ± 70.73	0.999
Using a PC, min/day	67.18 ± 80.25	72.25 ± 86.98	61.36 ± 71.70	0.290
Playing electronic games, min/day	50.96 ± 127.65	72.18 ± 162.68	26.58 ± 60.13	0.005
Using a smartphone, min/day	195.88 ± 203.25	177.40 ± 202.83	217.11 ± 202.53	0.128
Using tablets, min/day	19.45 ± 54.83	18.51 ± 58.34	20.53 ± 50.72	0.775
Studying, min/day	79.16 ± 90.30	64.39 ± 79.02	96.14 ± 99.41	0.006
Other activities, min/day	35.37 ± 72.58	28.82 ± 40.87	42.89 ± 96.67	0.130
Sedentary time on the weekend				
Total time, min/day	338.66 ± 226.45	335.50 ± 236.09	342.28 ± 215.82	0.816
Watching TV, min/day	98.27 ± 130.08	99.31 ± 142.46	97.06 ± 114.83	0.893
Using a PC, min/day	113.73 ± 149.31	133.63 ± 160.73	90.88 ± 132.03	0.025
Playing electronic games, min/day	89.10 ± 188.42	130.31 ± 238.01	41.75 ± 85.21	<0.001
Using a smartphone, min/day	261.99 ± 282.68	249.16 ± 318.50	276.74 ± 235.44	0.447
Using tablets, min/day	23.41 ± 68.64	23.93 ± 77.44	22.81 ± 57.20	0.899
Studying, min/day	89.31 ± 91.84	74.50 ± 96.69	106.32 ± 83.15	0.007
Other activities, min/day	39.51 ± 55.25	32.21 ± 44.52	47.89 ± 64.64	0.026
Emotional well-being	11.34 ± 2.83	11.94 ± 2.53	10.66 ± 2.99	<0.001
Social well-being	14.22 ± 5.75	15.48 ± 5.74	12.77 ± 5.43	<0.001
Psychological well-being	19.31 ± 6.07	20.55 ± 6.04	17.88 ± 5.82	<0.001
Physical well-being	17.90 ± 4.00	19.38 ± 3.52	16.19 ± 3.85	<0.001

**Table 2 nutrients-16-03827-t002:** Pearson correlations between the main study variables for the total sample of adolescents.

Variables	NOVA—UPF 24 h Recall				NOVA—UPF Away from Home			
	DY	PFF	SSS	Total	DY	PFF	SSS	Total
Body mass index	−0.06	−0.07	−0.08	−0.09	−0.08	−0.13 *	−0.11	−0.12
Body fat	−0.13 *	−0.10	−0.10	−0.14 *	−0.14 *	−0.12	−0.13 *	−0.14 *
Sedentary time during weekdays								
Total time	0.09	0.10	0.03	0.10	0.11	0.09	0.11	0.11
Watching TV	0.11	0.16 *	0.02	0.13 *	0.11	0.10	0.09	0.11
Using a PC	0.00	−0.03	0.05	0.01	0.09	0.05	0.07	0.07
Playing electronic games	0.18 **	0.23 **	0.05	0.20 **	0.05	−0.05	−0.02	−0.02
Using a smartphone	0.21 **	0.25 **	0.11	0.24 **	0.12	0.08	0.09	0.10
Using tablets	0.01	0.01	0.07	0.04	−0.05	−0.01	0.01	−0.01
Studying	−0.13 *	−0.18 **	0.00	−0.13 *	−0.08	−0.03	−0.07	−0.06
Other activities	−0.07	−0.03	−0.02	−0.05	−0.01	−0.03	−0.01	−0.02
Sedentary time during weekend days								
Total time	0.10	0.12	0.03	0.11	0.13 *	0.02	0.08	0.08
Watching TV	0.13 *	0.29 **	−0.01	0.18 **	0.10	0.08	0.07	0.09
Using a PC	−0.09	−0.03	0.00	−0.05	0.04	−0.01	0.02	0.01
Playing electronic games	0.22 **	0.28 **	−0.01	0.21 **	0.03	−0.02	−0.05	−0.02
Using a smartphone	0.06	0.13 *	0.05	0.10	0.00	0.02	0.05	0.03
Using tablets	−0.02	0.00	0.07	0.03	−0.03	0.01	0.00	−0.01
Studying	−0.18 **	−0.17 **	−0.02	−0.16 *	−0.13 *	−0.07	−0.06	−0.09
Other activities	−0.08	0.05	−0.01	−0.01	0.01	−0.01	0.01	0.00
Emotional well-being	0.02	−0.01	−0.02	−0.01	−0.03	0.01	−0.01	−0.01
Social well-being	0.11	0.04	0.03	0.07	0.07	0.12	0.10	0.11
Psychological well-being	0.08	0.06	0.04	0.08	0.05	0.10	0.07	0.08
Physical well-being	0.05	−0.01	−0.07	−0.02	0.04	0.06	0.00	0.04

Note: DY = drinks and yogurts; PFF = packaged and fast food; SSS = sweet and salty snacks; total = total UPF NOVA score; * *p* < 0.05; ** *p* < 0.01.

**Table 3 nutrients-16-03827-t003:** The association between UPF (previous day’s total score) and overweight risk, controlling for confounders (i.e., age, sex, sedentary behaviours, parental BMI, and parental education) in adolescents aged 12–17 years.

		Overweight/Obesity				
*n*	Model ^a^	*B*	S.E.	*e^B^*	95% C.I.	*p*
	1	−0.069	0.041	0.93	0.86 to 1.01	0.088
	2	−0.071	0.041	0.93	0.86 to 1.01	0.083
245	3	−0.093	0.050	0.91	0.83 to 1.01	0.063
	4	−0.079	0.052	0.92	0.84 to 1.02	0.130
	5	−0.094	0.053	0.91	0.82 to 1.01	0.076

^a^ Model 1 = unadjusted; Model 2 = model 1 + sex and chronological age; Model 3 = model 2 + sedentary behaviour on weekdays and at the weekend (i.e., TV viewing, PC, electronic games, iPads/tablets, smartphone) + active behaviour; Model 4 = model 3 + parental BMI; Model 5 = model 4 + parental education.

**Table 4 nutrients-16-03827-t004:** The association between UPF (away-from-home total score) and overweight risk, controlling for confounders (i.e., age, sex, sedentary behaviours, parental BMI, and parental education) in adolescents aged 12–17 years.

		Overweight/Obesity				
*n*	Model ^a^	*B*	S.E.	*e^B^*	95% C.I.	*p*
	1	−0.045	0.036	0.96	0.89 to 1.03	0.214
	2	−0.041	0.036	0.96	0.89 to 1.03	0.258
245	3	−0.053	0.043	0.95	0.87 to 1.03	0.222
	4	−0.045	0.043	0.96	0.88 to 1.04	0.300
	5	−0.078	0.047	0.98	0.84 to 1.02	0.098

^a^ Model 1 = unadjusted; Model 2 = model 1 + sex and chronological age; Model 3 = model 2 + sedentary behaviour on weekdays and at the weekend (i.e., TV viewing, PC, electronic games, iPads/Tablets, smartphone) + active behaviour; Model 4 = model 3 + parental BMI; Model 5 = model 4 + parental education.

## Data Availability

The data presented in this study are available upon request from the corresponding author due to privacy and ethical restrictions.

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
