# Peer review of "Ultra-Processed Food Consumption and Its Association with Risk of Obesity, Sedentary Behaviors, and Well-Being in Adolescents"

_nutrients, 2024, doi:10.3390/nu16223827_

Round 1
Reviewer 1 Report (New Reviewer)
Comments and Suggestions for Authors
Dear authors,
The study presents an epidemiological investigation on Diet, Sedentary behaviors and Well-Being, focused on Processed foods in Portuguese adolescents. Your research meets all methodological guarantees. But It is recommended to add more References with similar European epidemiological studies. And finally and mainly in the Discussion section, given that this Portugues adolescent population has compulsory education, it is very necessary some proposals for intervention and resources must be added for the promotion and education of Healthy Eating Habits, such as the Healthy Mediterranean Diet Pyramid and through the active teaching methodology with the Serious Games like Video Games or Educational Robotics.
Author Response
2024 November, 4th
Manuscript ID: NUTRIENTS-3229173 entitled " Ultra-processed food consumption and its association with risk of obesity, sedentary behaviors and well-being in Portuguese adolescents"
The authors thank the reviewers for their thoughtful and construction comments and suggestions. The comments were carefully read, and the revised manuscript was performed accordingly. Responses to specific comments are indicated below in BLUE font.
REVIEWER #1
REVIEWER #1:
Dear authors,
The study presents an epidemiological investigation on Diet, Sedentary behaviors and Well-Being, focused on Processed foods in Portuguese adolescents. Your research meets all methodological guarantees. But It is recommended to add more References with similar European epidemiological studies.
AUTHORS:
The authors really would like to thank the reviewer for their thoughtful and construction comments and suggestions. Even considering the lack of studies in the European context and particularly in Portugal, the authors tried to incorporate the couple of studies on this scientific field (UPF and lifestyle and/or risk of overweight in European adolescents). Therefore, the manuscript was edited including an additional reference (Ruggiero et al., 2021).
The present study achieves greater insight since there is very few studies in European youth. The authors are aware that there are several other studies but in adults; among adolescents there is several studies in south of America. Furthermore, there is no studies in adolescents from the south of Europe that investigate the combined association between UPF consumption and several behavioral (i.e. physical activity, sedentary behaviors) and emotional factors and the risk of overweight like the present study do.
Ruggiero E, Esposito S, Costanzo S, Di Castelnuovo A, Cerletti C, Donati MB, de Gaetano G, Iacoviello L, Bonaccio M; INHES Study Investigators. Ultra-processed food consumption and its correlates among Italian children, adolescents and adults from the Italian Nutrition & Health Survey (INHES) cohort study. Public Health Nutr. 2021 Dec;24(18):6258-6271. doi: 10.1017/S1368980021002767. Epub 2021 Jun 24. PMID: 34289922; PMCID: PMC11148574.
REVIEWER #1:
And finally and mainly in the Discussion section, given that this Portuguese adolescent population has compulsory education, it is very necessary some proposals for intervention and resources must be added for the promotion and education of Healthy Eating Habits, such as the Healthy Mediterranean Diet Pyramid and through the active teaching methodology with the Serious Games like Video Games or Educational Robotics.
AUTHORS:
Very good point; the authors really thank and agree with the reviewer. Therefore, according to the suggestion, the proposals for intervention were added to the manuscript providing a clearer and better focus on the discussion/conclusions.
That content was added as follows:
Furthermore, the results of the current study also highlight the need to encourage the improvement of the healthy nutritional habits in adolescents at school setting. For example, among the potential interventions to promote and educate about healthy eating habits are the construction and cultivation of school gardens, the development of recipes with students to use fresh produce from the garden while excluding ultra-processed ingredients, and the organization of healthy recipe competitions. Additionally, it is essential to implement actions and playful activities that encourage the availability of nutritious foods and promote healthy choices in school cafeterias, using tools such as the Food Wheel, the Mediterranean Diet Pyramid, and the NOVA classification as references.
Reviewer 2 Report (New Reviewer)
Comments and Suggestions for Authors
1.Overweight and obese were decided by BMI or other ways? Please define them in the manuscript.
2.Lines 260-262, “After controlling for all potential confounders, UPF tend to be associated with the risk of being overweight on the last day and away from home (p=
0.08).” why only one p-value? Please be consistent with Table 3 and Table 4.
3.Lines 262-264, “after controlling for sedentary and active behaviors (i.e. Model 3), adolescents who consumed UPF tend to be associated with higher risk of being overweight, but also marginally without statistical significance (p=0.06).” Please add “previous day total score” or the like.
4.Lines 276-279, It is better to have Tables in supplementary materials. Besides, for “83% less likely to be classified as overweight or obese youth,” it is better also to report the “eB value” in this place to help readers understanding the meaning.
5.For Table3 and Table 4, the below denoted “…sedentary behaviour on weekdays and at the weekend.” Were these two scores (“weekdays” and “at the weekend”) summed to produce a total score in the regression or other way? Please elucidate it. It is appreciated to have the detailed analysis outcome with Table in the supplementary materials.
6.In abstract, “both boys and girls of mothers with high educational level were less likely to be classified as overweight or obese youth.” Since it is a important finding, the analysis outcome with adequate Table would be presented in the content.
7.In abstract, the second question is “to investigate the association between the UPF consumption and risk of overweight, sedentary behaviors, and well-being in adolescents.” But why no report of answer on the question ? Such that it seemed to be obtained from Table 2 that no significant association between UPF and well-being.
8.Below the Table S1, please denote that boys’ score minus girls’ score for “mean difference.” Besides, for some numerical data it is better to use dot than comma.
Author Response
2024 November, 4th
Manuscript ID: NUTRIENTS-3229173 entitled " Ultra-processed food consumption and its association with risk of obesity, sedentary behaviors and well-being in Portuguese adolescents"
The authors thank the reviewers for their thoughtful and construction comments and suggestions. The comments were carefully read, and the revised manuscript was performed accordingly. Responses to specific comments are indicated below in BLUE font.
REVIEWER #2
REVIEWER #2:
1.Overweight and obese were decided by BMI or other ways? Please define them in the manuscript.
AUTHORS:
The authors thank and agree with the reviewer that a clarification is needed throughout the methods (i.e. Anthropometry). Therefore, according to the suggestion, that sub-heading of the manuscript was edited accordingly.
REVIEWER #2:
2.Lines 260-262, “After controlling for all potential confounders, UPF tend to be associated with the risk of being overweight on the last day and away from home (p= 0.08).” why only one p-value? Please be consistent with Table 3 and Table 4.
AUTHORS:
Done. The authors really thank and agree with the reviewer that results were somewhat incomplete through the results section. Therefore, specific details of the results were added to provide better understanding of the potential reader.
REVIEWER #2:
3.Lines 262-264, “after controlling for sedentary and active behaviors (i.e. Model 3), adolescents who consumed UPF tend to be associated with higher risk of being overweight, but also marginally without statistical significance (p=0.06).” Please add “previous day total score” or the like.
AUTHORS:
Thank you very much indeed for the careful reading and suggestion about the results, which provides its better contextualization. Thus, that content was incorporated.
REVIEWER #2:
4.Lines 276-279, It is better to have Tables in supplementary materials. Besides, for “83% less likely to be classified as overweight or obese youth,” it is better also to report the “eB value” in this place to help readers understanding the meaning.
AUTHORS:
Done. The authors thank and agree with the reviewer; thus, further details of results were added, especially the Odds ratio and Confidence Interval, as well as the level of significance on the association between those constructs/variables.
Thus, the authors have carefully read and double-checked the results section and have revised the manuscript accordingly.
REVIEWER #2:
5.For Table3 and Table 4, the below denoted “…sedentary behaviour on weekdays and at the weekend.” Were these two scores (“weekdays” and “at the weekend”) summed to produce a total score in the regression or other way? Please elucidate it. It is appreciated to have the detailed analysis outcome with Table in the supplementary materials.
AUTHORS:
Good point. The authors have clarified on the methods section (i.e. 2.4. Sedentary behaviors; 2.7. 2.7. Statistical Analyses) how sedentary behaviors were used on the different analyses to better elucidate the potential reader.
REVIEWER #2:
6.In abstract, “both boys and girls of mothers with high educational level were less likely to be classified as overweight or obese youth.” Since it is a important finding, the analysis outcome with adequate Table would be presented in the content.
AUTHORS:
The authors truly understand the reviewer’s concern. Once that result is important, but it is not the main result of the study, the authors decided to keep the Tables 3 and Table 4 with that clearer content/data. However, we follow the reviewer suggestion by added that important result in the text of the paper, with more detailed information (i.e. Odds ratio and Confidence Interval, as well as the level of significance on the association between those constructs/variables). Thank you.
REVIEWER #2:
7.In abstract, the second question is “to investigate the association between the UPF consumption and risk of overweight, sedentary behaviors, and well-being in adolescents.” But why no report of answer on the question ? Such that it seemed to be obtained from Table 2 that no significant association between UPF and well-being.
AUTHORS:
Thank you very much for the careful reading of the manuscript. In fact, the authors have answered the question, but we have chosen the main result to include on the abstract providing a more pragmatic message/”go home message”; in addition, it was cumulatively considered the editorial limitations of words’ count of that section. Furthermore, once the association between UPF and well-being of adolescents was not significant, the authors have decided to do not highlight it; thus, hopefully the reviewer’s understanding, we have focused on the main significant association between the other constructs (i.e. sedentary behaviors, and BMI/weight-status).
REVIEWER #2:
8.Below the Table S1, please denote that boys’ score minus girls’ score for “mean difference.” Besides, for some numerical data it is better to use dot than comma.
AUTHORS:
The authors also thank the reviewer for his careful reading and suggestions which were considered and indubitably contribute to make the manuscript clearer. Furthermore, the paper was edited by incorporating the reviewer’s suggestion on the Table S1.
This manuscript is a resubmission of an earlier submission. The following is a list of the peer review reports and author responses from that submission.
Round 1
Reviewer 1 Report
Comments and Suggestions for Authors
It is my contention that the study does not present a significant scientific report; indeed, it even contradicts published reports. The study sample is not representative of the population under investigation, which is a limitation of the study.
Abstract
It is possible that other factors may have influenced this conclusion, such as the high-calorie diets and low levels of physical activity observed among the remaining study participants.
Given that the statistical analysis did not indicate any correlations that could be attributed to the small study sample or lack of precision, a more detailed diagnosis should have been conducted. It is evident that greater exposure to ultra-processed foods is associated with an elevated risk of adverse health effects. Therefore, it would have been prudent to include and assess the relevant health indicators.
Furthermore, the article would have benefited from a more thorough examination of the sentences contained in lines 280-285.
Comments on the Quality of English LanguageThe article is beautifully described but it doesn't contribute anything
best regards
Author Response
The authors thank the reviewers for their thoughtful and construction comments and suggestions. The comments were carefully read, and the revised manuscript was performed accordingly. Responses to specific comments are indicated below in BLUE font.
REVIEWER #1
REVIEWER #1:
It is my contention that the study does not present a significant scientific report; indeed, it even contradicts published reports. The study sample is not representative of the population under investigation, which is a limitation of the study.
AUTHORS:
The authors agree with the reviewer that the sample of the present study is not representative of the population, particularly the Portuguese adolescent population, which could be a limitation. Therefore, the manuscript was edited recognizing that additional limitation, as follows: firstly, a better focus was made on the writing by correcting the title (i.e. removing the “Portuguese adolescents”), refining the research question and the hypotheses of the present study, as well as further terminological details on the methods and conclusion of the study.
Furthermore, to dissipate potential doubts about the sample dimension and its impact on the results/conclusion of the paper, a new and novel statistical analysis was run; thus, the analysis of the additional bootstrap estimates showed a similar trend of results (see Tables S1 and S2 in the Supplementary Materials).
On the other hand, as the reviewer clearly knows science is a rigorous, systematic endeavor that constructs and organizes knowledge in the form of testable explanations and predictions about our surroundings, according with our scientific experience, knowledge and scientific interests. Thus, one of the main drivers of the advancement of science is human diversity and curiosity, uncommitted to concrete results and free from any type of tutelage or rigid standard statement.
Thus, according to the existing literature on this specific field, as well as taking into account the scientific experience of our research team, the present study started to overcome the research gap by combined in the same analytical approach a large rage of variables from different dimensions and nature (i.e. behavioral, biological, and socio-cultural indicators). In addition, its design congregates several robust statistical procedures from comparative analysis to multi-method association tools (i.e. bivariate/partial correlations, logistic regression analyses controlling for several studies confounders).
The authors are completely aware, however, that no study exists without limitations which need to be recognized. Thus, the manuscript was refined accordingly by also emphasizing their strength of study related to the understudied population of the Southern Europe with high rates of overweight.
REVIEWER #1:
Abstract
It is possible that other factors may have influenced this conclusion, such as the high-calorie diets and low levels of physical activity observed among the remaining study participants.
AUTHORS:
The authors are completely in agreement with the reviewer that there are other factors that might impact on the association between the studied constructs. The hypercaloric diets and the levels of physical activity in its different levels of intensities are just a couple of examples among several others.
Thus, considering the higher interest to improve our manuscript, we have confirmed again all statistical procedures, and the active behaviors/physical activity was included as co-variable in the logistic regression analyses (i.e. model 3), as suggested by the reviewer; of note, it was in that model 3 controlling for related behavioral variables of energy expenditure that association between UPF and the risk of obesity was stronger but marginally without statistical significance (p=0.06). In addition, this important detail was also discussed at the manuscript; thus, at the limitations of the study the authors highlighted the assessment tools problematic issues by suggested objective measures of physical activity for further research, as well as the high-calorie diet which was not included in the present studied.
Of interest, the nutritional tools are also not without of weaknesses; for example, the used questionnaire by asking about the UPF of the previous day instead of the last week might be introducing additional bias. In fact, the combined factors of data collection, particularly the balance between the reliability of used measures and the ability of participants to report what are being measured (i.e. UPF), plays an important role on the experimental planning which need to be always considered in the future scientific challenges.
REVIEWER #1:
Given that the statistical analysis did not indicate any correlations that could be attributed to the small study sample or lack of precision, a more detailed diagnosis should have been conducted. It is evident that greater exposure to ultra-processed foods is associated with an elevated risk of adverse health effects. Therefore, it would have been prudent to include and assess the relevant health indicators.
AUTHORS:
The authors agree that ultra-processed food might be somewhat associated with adverse health effects. However, the authors think that general statement should be contextualized and specified according to which health factors we are referring to. Thus, following the afore-mentioned re-ran statistical procedures, a better focus was done using the variables included in present project; the results and conclusion were also refined accordingly.
The authors also thank the reviewer for his interesting reflection which indubitably contribute to improve the manuscript; thus, the paper was edited by incorporating also further suggestion to future research in the health-related adverse effects of ultra-processed food.
REVIEWER #1:
Furthermore, the article would have benefited from a more thorough examination of the sentences contained in lines 280-285.
AUTHORS:
The authors thank the reviewer for his careful reding of our manuscript, and particularly to advise us for better focus on the conclusions. Those sentences were rewritten according to the main results of the study.
REVIEWER #1:
Comments on the Quality of English Language
The article is beautifully described but it doesn't contribute anything.
AUTHORS:
According to the quality of the writing and related terminology, the authors truly thank this reviewer as well as the other reviewers for their constructive and coherent comments on the writing analysis.
On the other hand, the authors believe that complementary feature of the manuscript such as a well written manuscript could also contribute as good example for future quality of the research in this specific scientific field.
Reviewer 2 Report
Comments and Suggestions for Authors
Dear Authors,
The paper deals with important research area of Ultra-Processed Food in young people, and adolescents’ overweight, obesity, sedentary lifestyle, and well-being. The paper contributes to the knowledge needed for preventive activities in the field of child and young people health promotion and their healthy nutrition. In recent decades, the usage of “convenient” food, UPF, fast food is as growing epidemic for children in Europe, and as well in other continents. Therefore, this innovative, well-done study is of importance for the research and for general audience. The purpose of the study were:
-to compare the Ultra-Processed Food consumption, sedentary behaviors and well-being perception between boys and girls;
-to investigate the association between the Ultra-Processed Food consumption and risk of overweight, sedentary behaviors, and well-being in adolescents.
In order to improve the manuscript, I have a few comments and suggestions to the authors before the manuscript could be considered for publication. Please, find below my comments for the paper.
In Abstract
1. Please, do not use in the abstract unexplained abbreviations, for instance, PC.
2. The text in Conclusions could be corrected. Look bellow for the comment for Conclusions section.
In fact, the abstract could be as a plain text, without subtitles. In your case, maybe it is better to not separate Results and Conclusions.
Keywords
3. Instead of “sedentary time”, it is recommended to use “sedentary behaviors” or “sedentary lifestyle”.
Materials and Methods
The section in detail describes the methods, tools and variables used in the study.
Results
The findings of the study are presented in visible tables, and understandably text.
Discussion
The Discussion section is really well done, interesting and clear for the audience.
4. In Conclusions, the main achievements of the study are missing. It is recommended shortly to add here the information about differences by gender and mother education in words. Better not to use the percentages (%). Please, correct similarly in the Abstract.
General comment
5. In the paragraph it can not be the only one sentence. Please, combine and join the sentences (for instance, in the line 202-203, 242). Please, check throughout the text.
I hope my comments would be helpful for improvement of the manuscript.
Reviewer’s decision: Accept after revision
Author Response
REVIEWER #2:
REVIEWER #2:
Dear Authors,
The paper deals with important research area of Ultra-Processed Food in young people, and adolescents’ overweight, obesity, sedentary lifestyle, and well-being. The paper contributes to the knowledge needed for preventive activities in the field of child and young people health promotion and their healthy nutrition. In recent decades, the usage of “convenient” food, UPF, fast food is as growing epidemic for children in Europe, and as well in other continents. Therefore, this innovative, well-done study is of importance for the research and for general audience. The purpose of the study were:
-to compare the Ultra-Processed Food consumption, sedentary behaviors and well-being perception between boys and girls;
-to investigate the association between the Ultra-Processed Food consumption and risk of overweight, sedentary behaviors, and well-being in adolescents.
AUTHORS:
The authors really would like to thank the reviewer for their thoughtful and construction comments and suggestions. The comments were carefully read, and the revised manuscript was performed accordingly.
REVIEWER #2:
In order to improve the manuscript, I have a few comments and suggestions to the authors before the manuscript could be considered for publication. Please, find below my comments for the paper.
In Abstract
- Please, do not use in the abstract unexplained abbreviations, for instance, PC.
- The text in Conclusions could be corrected. Look bellow for the comment for Conclusions section.
In fact, the abstract could be as a plain text, without subtitles. In your case, maybe it is better to not separate Results and Conclusions.
AUTHORS:
Thank you for your suggestion and the abbreviations, as well as the conclusion were corrected/refined; however, for editorial reasons the authors decided to keep the subtitles. In addition, we have also incorporate part of your suggestions by adding information about sex differences and mother education issues.
REVIEWER #2:
Keywords
- Instead of “sedentary time”, it is recommended to use “sedentary behaviors” or “sedentary lifestyle”.
AUTHORS:
Done.
REVIEWER #2:
Materials and Methods
The section in detail describes the methods, tools and variables used in the study.
AUTHORS:
Thank you very much for the constructive comments.
REVIEWER #2:
Results
The findings of the study are presented in visible tables, and understandably text.
AUTHORS:
Thank you.
REVIEWER #2:
Discussion
The Discussion section is really well done, interesting and clear for the audience.
AUTHORS:
Thank you very much indeed for the experienced comments.
REVIEWER #2:
- In Conclusions, the main achievements of the study are missing. It is recommended shortly to add here the information about differences by gender and mother education in words. Better not to use the percentages (%). Please, correct similarly in the Abstract.
AUTHORS:
The authors agree somewhat with the reviewer, and the conclusion was refined accordingly.
REVIEWER #2:
General comment
- In the paragraph it can not be the only one sentence. Please, combine and join the sentences (for instance, in the line 202-203, 242). Please, check throughout the text.
AUTHORS:
The authors thank the reviewer for his/her careful reading, and those paragraphs were refined accordingly.
REVIEWER #2:
I hope my comments would be helpful for improvement of the manuscript.
Reviewer’s decision: Accept after revision
AUTHORS:
The authors thank the reviewer for his/her constructive comments which contributed for the improvement of our manuscript.
Reviewer 3 Report
Comments and Suggestions for Authors
Peer review report for the article: "Ultra-processed food consumption and its association with risk of obesity, sedentary behaviors and well-being in Portuguese adolescents"
The article addresses the significant public health concern of paediatric obesity, focusing on the role of ultra-processed food (UPF) consumption, sedentary behavior, and well-being among Portuguese adolescents. The study is well-structured, uses appropriate methods, and offers useful insights. However, there are several areas where clarification, expansion, or additional analysis is needed to strengthen the manuscript.
Comments:
1. While the abstract effectively summarizes the study, it could benefit from a clearer description of the non-significant findings related to UPF and obesity. For example, it mentions "UPF was not significantly associated with the risk of being overweight" but does not explain this outcome. Adding a sentence about potential reasons or contextualizing this result within the broader literature would improve clarity
2. The introduction provides a solid overview of UPF and its association with obesity and mental health outcomes. However, it would benefit from a more explicit focus on the specific gaps this study is addressing, particularly in relation to sedentary behaviors and well-being in the Portuguese context.
3. The hypotheses are stated at the end of the introduction, which is appropriate, but they could be more explicitly linked to the gaps in existing research. Additionally, while the study hypothesizes that UPF will be related to obesity, it doesn’t explicitly hypothesize about the maternal education variable or sedentary behaviors, which later emerge as significant.
4. The description of the sampling design indicates that the study is an extension of previous projects. However, it does not clearly explain how the participants were recruited or selected from the two schools in Coimbra and Viseu. More detail is necessary to understand the representativeness of the sample
5. The age range of 12-17 years is standard, but it may help to include a brief justification for focusing on this specific range in the context of adolescent development and health behaviors.
6. Ultra-Processed Food Consumption- The scoring system (0 to 28) is mentioned, but it could be beneficial to explain what constitutes a high or low score in the context of this study. Include thresholds for low, moderate, and high consumption of UPF according to the scoring system, if available.
7. Imputation Methodology: The method for handling missing data is stated as the expectation-maximization algorithm, which is a good choice. However, the rationale for selecting this method could be elaborated upon.
Author Response
REVIEWER #3:
REVIEWER #3:
Peer review report for the article: "Ultra-processed food consumption and its association with risk of obesity, sedentary behaviors and well-being in Portuguese adolescents"
The article addresses the significant public health concern of paediatric obesity, focusing on the role of ultra-processed food (UPF) consumption, sedentary behavior, and well-being among Portuguese adolescents. The study is well-structured, uses appropriate methods, and offers useful insights. However, there are several areas where clarification, expansion, or additional analysis is needed to strengthen the manuscript.
Comments:
- While the abstract effectively summarizes the study, it could benefit from a clearer description of the non-significant findings related to UPF and obesity. For example, it mentions "UPF was not significantly associated with the risk of being overweight" but does not explain this outcome. Adding a sentence about potential reasons or contextualizing this result within the broader literature would improve clarity
AUTHORS:
The authors thank and agree with the reviewer that a better contextualization is needed throughout the paper. Therefore, according to the suggestion, that part of the abstract was rewritten to provide a clearer and better focus on the conclusions, as well as throughout the discussion/limitations of the paper (i.e. highlighting the assessment tools problematic issues and specific socio-cultural features of communities where data were collected).
REVIEWER #3:
- The introduction provides a solid overview of UPF and its association with obesity and mental health outcomes. However, it would benefit from a more explicit focus on the specific gaps this study is addressing, particularly in relation to sedentary behaviors and well-being in the Portuguese context.
AUTHORS:
Considering the reviewer´s suggestion, a more pragmatic statement was provided on the relationship between studied constructs, particularly in the Portuguese context. In fact, in that national context there is no studies that analyse the afore-mentioned constructs together and/or combined (i.e. UPF and risk of obesity, sedentary time in different ways/types, and even on the different dimensions of well-being) among adolescents, so, claiming for further research.
REVIEWER #3:
- The hypotheses are stated at the end of the introduction, which is appropriate, but they could be more explicitly linked to the gaps in existing research. Additionally, while the study hypothesizes that UPF will be related to obesity, it doesn’t explicitly hypothesize about the maternal education variable or sedentary behaviours, which later emerge as significant.
AUTHORS:
Thank you for the note. The hypotheses of the manuscript were refined according to the reviewer´s suggestion.
REVIEWER #3:
- The description of the sampling design indicates that the study is an extension of previous projects. However, it does not clearly explain how the participants were recruited or selected from the two schools in Coimbra and Viseu. More detail is necessary to understand the representativeness of the sample.
AUTHORS:
Thank you for the note. Further details were added to the sampling design description.
The authors agree with the reviewer on his/her concern about the understanding of the representativeness of the sample. Thus, the authors would like to clarify that the sample of the present study is not representative of the population, particularly the Portuguese adolescent population, which could be a limitation. Therefore, the manuscript was edited recognizing that additional limitation, as follows: firstly, a better focus was made on the writing by correcting the title (i.e. removing the “Portuguese adolescents”), refining the research question and the hypotheses of the present study, as well as further terminological details on the methods and conclusion of the study.
Furthermore, to dissipate potential doubts about the sample dimension and its impact on the results/conclusion of the paper, a new and novel statistical analysis was run; thus, due to the moderate sample size (n< 250), bootstrap resampling methods (Field, 2013) were also used to estimate bias-corrected and accelerated (BCa) 95% confidence intervals for the mean differences between sex groups and for the Pearson’s correlation coefficients. Robust estimates were obtained based on 5000 bootstrap samples (random number generator: Mersenne Twister, seed 151024).
The afore-mentioned analysis of the additional bootstrap estimates showed a similar trend of results (see Tables S1 and S2 in the Supplementary Materials).
REVIEWER #3:
- The age range of 12-17 years is standard, but it may help to include a brief justification for focusing on this specific range in the context of adolescent development and health behaviors.
AUTHORS:
The authors agree with the reviewer that the age rage of 12-17 years is standard but, moreover, a brief justification and/or contextualization was added/included on the methods selection of the manuscript, as follows: “the sample age range is coincident of the scholar level or academic years studied at the same degree; since it was studied students from the elementary school (Portuguese 3rd cycle of the elementary studies), participants are aged from 12 to 17 years of age. According to the literature (Campbell et al., 2021), once gender is commonly indicated as factors affecting physical activity, well-being and weight status across adolescence, analyses were done separately for male and female adolescents.”
REVIEWER #3:
- Ultra-Processed Food Consumption- The scoring system (0 to 28) is mentioned, but it could be beneficial to explain what constitutes a high or low score in the context of this study. Include thresholds for low, moderate, and high consumption of UPF according to the scoring system, if available.
AUTHORS:
Very good point. Considering the reviewer’s suggestion, additional explanation and/or contextualization was included to the methods section of the paper by explaining the UPF score was analysed as a continuous variable following the original procedure used throughout the cited studies. In fact, there is no thresholds scoring system available using categorical variables, instead the continuous variable used in the present study provides more sensibility to the analytical approaches used in the present study.
REVIEWER #3:
- Imputation Methodology: The method for handling missing data is stated as the expectation-maximization algorithm, which is a good choice. However, the rationale for selecting this method could be elaborated upon.
AUTHORS:
Thanks for the note. Further details were provided to clarify the methodological procedure. Thus, the following content was also added to the manuscript: Missing data was imputed using the expectation-maximization algorithm based on all available data, since it has previously demonstrated to be an effective technique for small to moderate sample sizes (Malan et al., 2020).
Furthermore, due to the moderate sample size (n< 250), bootstrap resampling methods (Field, 2013) were also used to estimate bias-corrected and accelerated (BCa) 95% confidence intervals for the mean differences between sex groups and for the Pearson’s correlation coefficients. Robust estimates were obtained based on 5000 bootstrap samples (random number generator: Mersenne Twister, seed 151024).
Malan L, Smuts CM, Baumgartner J, Ricci C. Missing data imputation via the expectation-maximization algorithm can improve principal component analysis aimed at deriving biomarker profiles and dietary patterns. Nutr Res. 2020;75:67-76. doi:10.1016/j.nutres.2020.01.001.
Field A. Discovering statistics using IBM SPSS statistics. 4th ed. London, England: SAGE Publications; 2013.